# OptMaster: A DAG-Based Framework for Formulation and Heuristic Discovery in Optimization

Hang Lin [* 1 2]  Yuanpeng Gao [* 3 4]  Yuzhi Zhang [2 3]  Kun Yuan [4]  Gang Yan [1]  Siheng Chen [5 2]  Linfeng Zhang [3 6]
Weinan E [6 4]

## Abstract

Optimization problems are fundamental across science and industry, including planning, scheduling, and resource allocation. While LLMs show promise in automating optimization, they struggle to bridge the gap between real-world requirements and both mathematical formulations and effective heuristic designs. Furthermore, the field lacks a unified framework that spans problem formulation and heuristic discovery for NP-hard settings. To address these challenges, we propose **OptMaster**, a unified framework that spans optimization from formulation to heuristic discovery, structuring the process as a Directed Acyclic Graph (DAG) where each node represents a candidate solution. The DAG architecture enables cross-branch knowledge transfer when search progress stagnates. Within each node, we further replace textual self-reflection with independently generated verification code, grounding the evaluation in deterministic computation to suppress hallucinations. OptMaster achieves competitive performance across two optimization paradigms. In **Formulation Intelligence**, OptMaster achieves state-of-the-art accuracy across the three most challenging benchmarks in the field. In **Heuristic Discovery**, OptMaster matches or surpasses the best known solutions on Circle Packing ($n = 26, 32$) and achieves a cut of 9,590 on Gset70 with significantly reduced time and search budgets.

## 1. Introduction

Optimization plays a central role in decision-making across many domains of modern society. It supports vehicle routing in logistics networks (Gu et al., 2024), surgery scheduling in healthcare systems (Al Amin et al., 2025), crew assignment in airline operations (Xu et al., 2024), and production sequencing in manufacturing (Guzman et al., 2022). It is also central to scientific discovery, from ground-state search in quantum many-body systems (Chen & Heyl, 2024) to molecular optimization for drug design (Xia et al., 2024). Despite their diversity, these problems share a common form: optimizing an objective subject to constraints.

Recent advances in LLMs have increased interest in using them to build and solve optimization models from natural language descriptions. Existing work mainly follows two directions: formulation and heuristic discovery. Formulation methods use LLMs to construct mathematical models and generate solver-ready code, while heuristic discovery methods use LLMs to design and improve heuristics for hard optimization problems.

However, formulation methods often rely on natural-language self-verification, which can miss semantic errors and lead to incorrect models. Heuristic discovery methods often pass only a scalar score between iterations, which limits what later attempts can inherit. In both directions, solution attempts are treated independently, so intermediate results and partial insights are not systematically transferred across attempts.

In this paper, we propose OptMaster, a unified LLM-based optimization framework that integrates problem formulation and heuristic discovery through a knowledge-sharing exploration process. Given a natural language problem description, along with domain constraints and raw data, OptMaster constructs a Directed Acyclic Graph (DAG) of solution attempts, where each node corresponds to a complete, executable optimization strategy. At each node, the agent generates solver code, executes it using external tools and compute resources, and evaluates the resulting solution with independent verification code. Crucially, execution outcomes and verified intermediate insights, such as feasi-

---

[*]Equal contribution  [1]Tongji University, Shanghai, China [2]Sciland Technology, Shanghai, China [3]DP Technology, Beijing, China [4]Peking University, Beijing, China [5]Shanghai Jiao Tong University, Shanghai, China [6]AI for Science Institute, Beijing, China. Correspondence to: Linfeng Zhang <zhanglf@dp.tech>, Yuzhi Zhang <zhangyz@dp.tech>.

*Proceedings of the 43rd International Conference on Machine Learning*, Seoul, South Korea. PMLR 306, 2026. Copyright 2026 by the author(s).

bility signals, constraint violations, and algorithmic patterns are distilled into structured summaries that are transferred across nodes in the DAG.

Our contributions are:

- We propose OptMaster, a unified agent that addresses both formulation and heuristic discovery tasks in optimization within one framework.

- We propose a DAG-based search engine where each node is an executable attempt with an independent solve, verify, summarize lifecycle, enabling reliable evaluation and knowledge transfer across branches.

- We demonstrate strong performance across two settings: state-of-the-art accuracy on IndustryOR (87.3%), Mamo-Complex (95.6%), and OptMATH (66.9%) with minimal search overhead, and competitive results on heuristic discovery, matching or surpassing best known solutions on Circle Packing ($n$=26, 32) and achieving 9590 on MAXCUT Gset70 under reduced time and search budgets.

## 2. Related Work

**Large Language Models for Optimization Formulation.** For optimization formulation, research has expanded from early linear-programming focused efforts such as NL4Opt (Ramamonjison et al., 2022) to broader classes including mixed-integer, quadratic, and nonlinear programming. Benchmarks such as Mamo-Complex (Huang et al., 2025b), ComplexOR (Xiao et al., 2024), IndustryOR (Huang et al., 2025a), and OptMATH (Lu et al., 2025) provide standardized evaluations across these settings. Related work can be divided into prompt-based and learning-based methods. Recent prompt-based methods explore diverse modeling workflows. Some use multi-agent decomposition and coordination, such as OptiMUS (Ahmaditeshnizi et al., 2024), Chain-of-Experts (Xiao et al., 2024) and OptiTree (Liu et al., 2025). The learning-based methods aim to internalize optimization knowledge by updating the model parameters. LLMOPT (Jiang et al., 2025) uses mathematical five-element tuples to fine-tune the model. AutoFormulation (Astorga et al., 2025) adopts search-based formulation generation with pruning. There are also reinforcement learning approaches that refine models using solver feedback (Chen et al., 2025b; Zhou et al., 2026; Ding et al., 2026). Although these methods have demonstrated strong performance on existing benchmarks, a recent survey (Xiao et al., 2025) highlighted quality issues in current benchmark datasets.

**Large Language Models for Heuristic Discovery.** In parallel, for NP-hard settings where exact solvers do not scale, LLM-driven heuristic discovery has become a trend.

FunSearch (Romera-Paredes et al., 2024) showed that pairing an LLM with an evaluator enables evolutionary search over programmatic heuristics, which inspired follow-up systems such as EoH (Liu et al., 2024), ReEvo (Ye et al., 2024), and CALM (Huang et al., 2026). More recent frameworks scale this direction further. AlphaEvolve (Novikov et al., 2025) applies large-scale evolutionary search and achieves strong performance on several optimization tasks. ShinkaEvolve (Lange et al., 2026) improves sample efficiency via better parent selection, novelty-aware rejection sampling, and bandit-based ensemble selection. FM Agent (Li et al., 2025) uses population-based evolutionary optimization to search over heuristic designs and performs competitively on heuristic optimization benchmarks. These systems highlight the value of execution-based feedback for discovering effective algorithms.

## 3. Methodology

Figure 1 summarizes OptMaster as a three-part system. Exploration Planning explores a wide range of solution directions. Node Lifecycle executes a reusable solve, verify, and summarize loop to produce transferable evidence. Adaptive DAG Expansion increases recombination and multi-parent fusion under stagnation to restore diversity. We detail these components in §3.1, §3.2, and §3.3.

### 3.1. Exploration Planning: Diversifying Solution Trajectories

In LLM-based optimization, the agent often locks into one approach early. Later iterations mainly refine that choice. This limits exploration of other promising options. In real problems, the best approach is rarely obvious at the start. Different solving paradigms and modeling choices can lead to different outcomes.

OptMaster introduces an explicit exploration planning phase before search. This phase generates multiple distinct solution trajectories to encourage diverse exploration. The planner enumerates candidate trajectories and launches each as an independent branch. In practice, these trajectories often vary along two aspects: how to solve and what to solve. The former explores different solving strategies, such as solver-based formulations versus heuristic discovery. The latter explores different interpretations of the specification, such as objective priority, constraint strictness, and variable definitions.

This phase also supports human-in-the-loop guidance. Experts can edit the planned trajectories, add domain priors, and remove infeasible directions. This steers the search toward solutions that match real operational requirements.

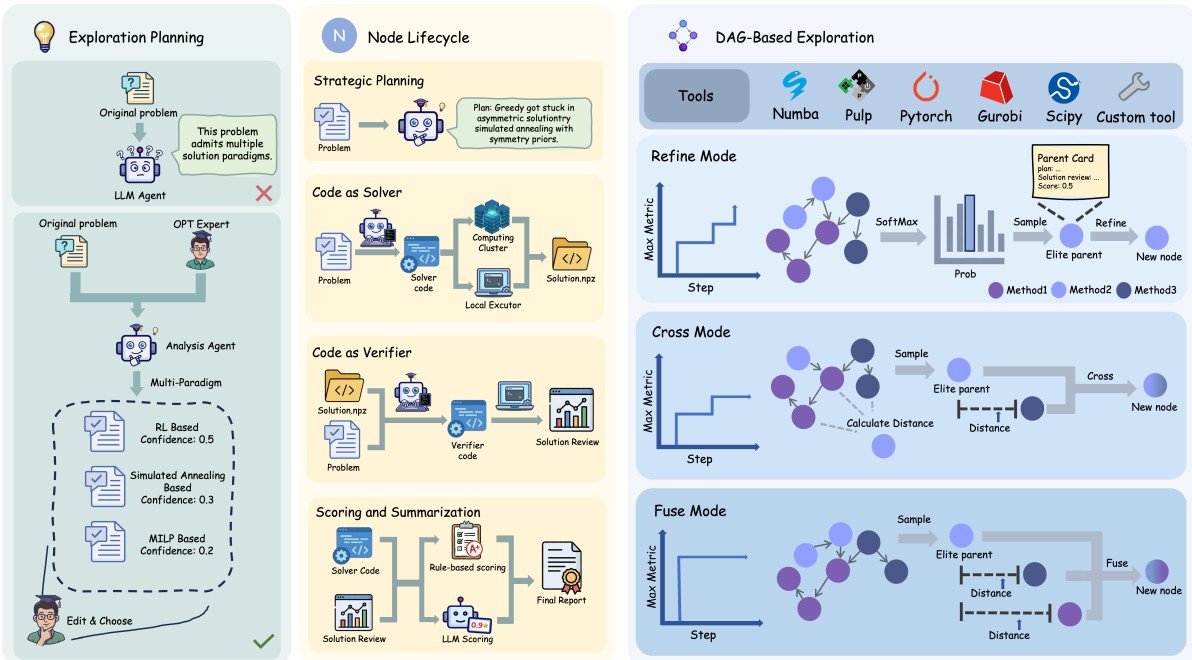

*Figure 1.* Overview of OptMaster. A unified framework for optimization that bridges natural language requirements and high-performance, verified solutions via a strategic DAG-based exploration engine

## 3.2. Node Lifecycle: Executable Solving and Verification

For each planned trajectory, OptMaster runs an iterative search over a DAG. Each node is a complete solving attempt, and edges transfer high-level information across attempts. We define the node state $\mathcal{S}_v$ as a full snapshot of attempt $v$:

$$\mathcal{S}_v = \langle P_v, C_v, A_v, V_v, \mathcal{R}_v \rangle. \tag{1}$$

Here, $P_v$ is the node-level plan, including modeling decisions and algorithmic choices. $C_v$ is executable code that implements $P_v$. $A_v$ is the persisted solution artifact produced by running $C_v$. $V_v$ is a deterministic verification report computed from $A_v$. $\mathcal{R}_v$ is a concise record of what was tried and what the verifier confirmed, and it is reused as transferable context by later nodes.

### 3.2.1. CODE AS SOLVER

Each node $v$ carries a node-level plan $P_v$. $P_v$ is generated by the planning module for this node, conditioned on the problem instance $\mathcal{I}$ and the inherited summaries from its parent nodes. It specifies the intended solving approach, including key modeling decisions and algorithmic choices. OptMaster instantiates $P_v$ as executable solver code $C_v$ and runs it to obtain a candidate solution.

$$(P_v, C_v) = \text{LLM}\big(\mathcal{I}, \{T_u\}_{u \in \text{pa}(v)}\big). \tag{2}$$

Here $T_u$ denotes the transferable context passed from parent node $u$, which we define later in §3.3.

Many execution steps are shared across nodes, such as data loading, external solver or simulator calls, logging, and artifact saving. To avoid repeated and error-prone reimplementation, OptMaster provides these steps as reusable utilities $\mathcal{U}$. Node code focuses on the core solving logic, while $\mathcal{U}$ handles the shared infrastructure. Formally, we generate code from the plan and execute it to produce a persistent solution artifact:

$$A_v = \text{EXECUTE}(C_v, \mathcal{U}, b_v). \tag{3}$$

The backend flag $b_v$ selects the execution environment:

$$b_v \in \{\texttt{local}, \texttt{computing\_cluster}\}. \tag{4}$$

When $b_v = \texttt{computing\_cluster}$, heavy computation is offloaded to the computing cluster and the saved artifacts are collected after completion. We store $A_v$ as structured files rather than raw tokens. This keeps large solution objects out of the LLM context and makes downstream verification clean and reproducible.

### 3.2.2. CODE AS VERIFIER

Reliable evaluation is a major bottleneck in LLM-based optimization. Natural-language self-verification is unreliable and difficult to reuse. OptMaster therefore makes verification executable.

For each node, the system verifies the solution artifact $A_v$ against the problem instance $\mathcal{I}$. The verifier produces a

structured report $V_v$. This report is used for node selection and stored in the transferable summary $\mathcal{R}_v$.

The verifier code $C_v^{\mathrm{ver}}$ is generated for each node:

$$C_v^{\mathrm{ver}} = \mathrm{LLM}_{\mathrm{code}}(\mathcal{I}). \qquad (5)$$

A key design choice is strict separation between solving and verification. The verifier depends only on $\mathcal{I}$ and $A_v$. It does not read the solver code and does not access intermediate reasoning. All required quantities are recomputed directly from the artifact. This separation prevents generation errors from affecting evaluation. It also ensures reproducibility.

The generated verifier is then executed on the artifact:

$$V_v = \mathrm{EXECUTE}(C_v^{\mathrm{ver}}, \mathcal{U}, A_v). \qquad (6)$$

The same verifier supports both formulation and heuristic discovery. For formulation attempts, it checks feasibility and recomputes the objective under the original specification. For heuristic discovery attempts, it produces a diagnostic report that characterizes the current bottleneck. The report indicates where progress is limited and which components are responsible.

### 3.2.3. SCORING AND SUMMARIZATION

The lifecycle ends by converting verification into signals for the next step. Given the verifier report $V_v$, OptMaster produces a scalar reward $r_v$ for node ranking and a concise summary $\mathcal{R}_v$.

We score nodes in two ways. When the task provides a reliable metric, we compute $r_v$ directly from the verified quantities in $V_v$, such as feasibility and the recomputed objective. When no reliable metric is available, we use an LLM scorer that reads $\mathcal{I}$ and $V_v$ and outputs a normalized reward. This keeps rewards comparable across attempts within the same search run.

The summary $\mathcal{R}_v$ records only what later nodes need. It includes the node plan $P_v$ and the verified evidence in $V_v$. It excludes raw code and large artifacts to keep context small and stable.

Formally, we define:

$$r_v = \mathrm{SCORE}(\mathcal{I}, V_v), \qquad (7)$$

$$\mathcal{R}_v = \mathrm{SUMMARIZE}(P_v, C_v, V_v). \qquad (8)$$

### 3.3. DAG-Based Exploration

Having defined what each node contains, we now describe how nodes are connected and how the search expands. OptMaster organizes exploration as a directed acyclic graph (DAG), rather than a tree. A tree forces each node to inherit from a single parent, so branches evolve largely independently. In optimization, good solutions often combine partial gains from different branches. For example, one branch may discover a better constraint encoding, while another finds a stronger heuristic. The DAG allows a new node to inherit from multiple parents and combine these insights.

### 3.3.1. KNOWLEDGE TRANSFER BETWEEN NODES

Multi-parent inheritance needs a compact interface that can be reused. We do not pass solver code $C_v$ between nodes. Solver code is long and often mixed with instance-specific details. Copying code from multiple parents quickly bloats the context and is hard to combine. It also forces the new node to inherit unnecessary implementation details.

Instead, OptMaster transfers information at the strategy level. For optimization, the reusable part is not the exact implementation, but the decisions and verified evidence. This includes variable and constraint choices in the plan, and what the verifier confirmed from the produced artifact.

Following the node state in Eq. (1), we define the transferable context of a parent node $v$ as

$$T_v = \langle P_v, \ V_v, \ \mathcal{R}_v \rangle. \qquad (9)$$

We pass $T_v$ along DAG edges and use it to guide the generation of new nodes. $P_v$ provides the node strategy, $V_v$ provides verified evidence from execution, and $\mathcal{R}_v$ provides a concise record of what worked and what failed. This context is compact enough for multi-parent inheritance, while keeping the key information needed for later improvement.

### 3.3.2. STAGNATION-RESPONSIVE EXPANSION

Multi-parent inheritance expands the strategy space, but it should not be used all the time. When the search is still improving, we prefer single-parent refinement for efficiency. When progress slows, we increase cross-branch recombination to reintroduce diversity. We implement this idea with a stagnation signal and an adaptive parent sampling rule.

**Stagnation metric.** Let $r_t$ be the reward of the node created at step $t$. Define the best reward so far as $r_t^* = \max_{u \in \mathcal{V}} r_u$. We measure stagnation by the run length since the last improvement:

$$\tau_t = t - \max\{i \le t : r_i = r_t^*\}. \qquad (10)$$

We convert $\tau_t$ into a normalized stagnation score:

$$S_t = \min\left(\frac{\tau_t}{W}, \ 1 - \varepsilon\right), \qquad (11)$$

where $W$ is a patience window and $\varepsilon \in (0, 1)$ is a floor parameter. $S_t$ increases as the search keeps failing to improve. It saturates at $1 - \varepsilon$. This guarantees that refinement keeps probability at least $\varepsilon$ even under long stagnation.

**Sampling the number of parents.** We choose the number of parents $k \in \{1, 2, 3\}$ using a sequential inclusion process. We start from one parent. We add a second parent with probability $S_t$. Conditioned on having two parents, we add a third parent with probability $S_t$. This yields a truncated geometric distribution:

$$\Pr(k) = \begin{cases} 1 - S_t, & k = 1, \\ S_t(1 - S_t), & k = 2, \\ S_t^2, & k = 3. \end{cases} \quad (12)$$

This rule has two useful properties. First, $\Pr(k \geq 2) = S_t$, so recombination probability increases exactly with stagnation. Second, the expected parent count is bounded:

$$\mathbb{E}[k] = 1 + S_t + S_t^2 \leq 3. \quad (13)$$

This keeps the inherited context size stable.

These values define three stochastic expansion modes: Refine ($k = 1$), Cross ($k = 2$), and Fuse ($k = 3$). In addition, we use a deterministic Repair mode for execution failures. Repair is triggered when a selected parent has unresolved runtime errors. The repair node keeps the same high-level intent and fixes implementation issues.

**Parent selection.** Given $k$, we select parents with a quality-diversity trade-off.

*First parent (quality).* We sample the first parent from the global node pool $\mathcal{V}$ using a Boltzmann policy:

$$\Pr(p_1 = v) = \frac{\exp\left(\frac{r_v}{\beta}\right)}{\sum_{u \in \mathcal{V}} \exp\left(\frac{r_u}{\beta}\right)}. \quad (14)$$

where $\beta$ is a temperature parameter. This rule is the maximum-entropy distribution that prefers high-reward nodes. It avoids deterministically picking only the current best node.

*Additional parents (diversity).* For $j \geq 2$, we prefer parents that are both high quality and complementary to the already selected set. Let $\mathcal{P}$ denote the current parent set. We define a distance function $d(v, p)$ between two nodes by comparing their node-level plans. In practice, we embed each plan $P_v$ into a vector representation and compute cosine distance. We then define the novelty of a candidate node $v$ against $\mathcal{P}$ as

$$D(v, \mathcal{P}) = \min_{p \in \mathcal{P}} d(v, p). \quad (15)$$

This quantity is large only when $v$ is far from every selected parent, so it discourages redundant parents. We select $p_j$ by maximizing a scalarized quality-diversity objective:

$$p_j = \arg\max_{v \notin \mathcal{P}} \left[ r_v + S_t \cdot D(v, \mathcal{P}) \right]. \quad (16)$$

$S_t$ controls the quality-diversity trade-off: small $S_t$ favors reward, large $S_t$ favors diversity.

**Repair mode.** A node is *buggy* if execution fails due to runtime errors, timeouts, or invalid artifacts. Let $\mathcal{B}$ be the set of buggy nodes whose errors are not yet fixed. If $\mathcal{B}$ is nonempty, we repair before any further exploration by creating a child node conditioned on $T_v$. The child keeps the same solving strategy and only fixes the implementation to make the attempt runnable. If $\mathcal{B}$ is empty, we proceed with stagnation-responsive expansion using the sampling rule described above.

**Parallel Exploration.** At each step, multiple workers expand different parent sets in parallel. This increases coverage of the search space and reduces wall-clock time. Parallelism is orthogonal to the parent selection rule and does not change the search logic.

## 4. Experiments

We evaluate OptMaster on two settings: **Formulation Intelligence** (§4.1) and **Heuristic Discovery** (§4.2). On formulation benchmarks, OptMaster consistently improves over the backbone model. On heuristic discovery tasks, it discovers competitive executable heuristics under a small search budget and transfers across instance sizes.

### 4.1. Formulation Intelligence

#### 4.1.1. EXPERIMENTAL SETUP

**Datasets, Metrics, and Evaluation Protocol.** Experiments use three formulation benchmarks: IndustryOR (Huang et al., 2025a) (100 industrial scenarios), Mamo-Complex (Huang et al., 2025b) (203 undergraduate-level problems), and OptMATH (Lu et al., 2025) (166 problems across LP, MILP, NLP, and SOCP). Following OptMATH (Lu et al., 2025), a prediction is correct if the relative error between the predicted and ground-truth objective values is below $10^{-6}$. Results are averaged over three random seeds and reported as mean $\pm$ standard deviation. The benchmark test sets follow the public releases used by SIRL (Chen et al., 2025b) and LLMOPT (Jiang et al., 2025). OptMaster uses GPT-5 as the backbone model and introduces no expert priors or handcrafted tricks.

When the specification is clear, the planner generates a single interpretation; otherwise, it considers up to two interpretations. Each interpretation is searched as an independent branch, with at most three formulation attempts per branch. Each attempt corresponds to one node and receives a score from an LLM-based evaluator. The evaluator reads the original problem and the solver verification report, then outputs a score in $[0, 1]$. Search stops early when the score exceeds 0.94. We report two selection protocols. *Base* selects the single highest-scoring node among all candidates across all branches. *Two-Interpretations* selects the highest-

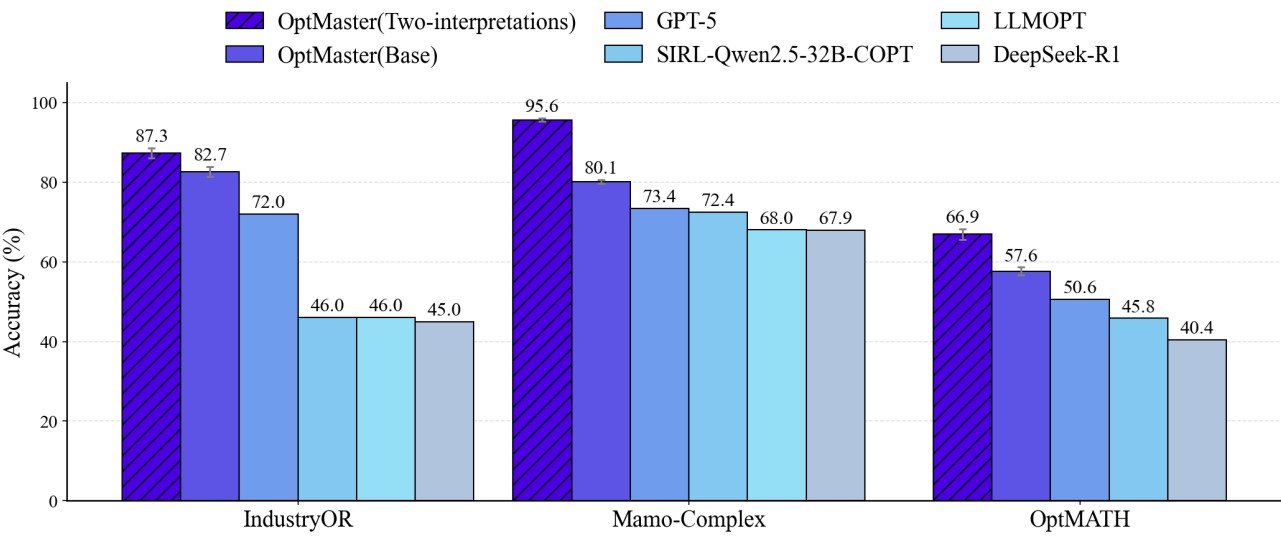

*Figure 2.* Formulation accuracy on three benchmarks (%, mean±std over 3 runs).

scoring node within each interpretation branch and marks an instance correct if either selected formulation matches the ground truth. We compare against frontier foundation models (GPT-5, DeepSeek-R1) and optimization-focused models (SIRL-Qwen2.5-32B-COPT (Chen et al., 2025b), LLMOPT (Jiang et al., 2025)). No external tools or knowledge sources are used, and Gurobi is used as the solver.

### 4.1.2. MAIN RESULTS

Figure 2 and Table 4 (in Appendix F) summarize results on the three main benchmarks. OptMaster achieves the best performance across all of them. OptMaster (Base) improves over GPT-5 by 10.7% on IndustryOR, 6.7% on Mamo-Complex, and 7.0% on OptMATH. OptMaster (Two-Interpretations) further improves accuracy to **87.3%**, **95.6%**, and **66.9%**. Based on the results, we find that general-purpose frontier models (e.g., GPT-5) now surpass specialized fine-tuned baselines due to their mathematical reasoning capabilities. OptMaster (Base) further elevates performance by introducing the Node Lifecycle, especially the *Code as Verifier* mechanism, which effectively mitigates hallucinations. OptMaster (Two-Interpretations) achieves the most significant gains. This highlights the necessity of *Exploration Planning* and *DAG-Based Exploration* in addressing the semantic ambiguity and complexity of real-world optimization as illustrated in our case study (Figure 7 in Appendix D).

### 4.2. Heuristic Discovery

Beyond formulation, many real-world optimization problems are NP-hard. Strong performance depends on effective heuristics rather than exact solvers. We therefore evaluate

OptMaster on two heuristic discovery tasks, **MAXCUT** on large graphs and **Circle Packing** with mixed continuous and discrete decisions.

### 4.2.1. EXPERIMENTAL SETUP

We use GPT-5 as the backbone LLM for all heuristic discovery experiments, with prompts and decoding settings fixed across runs. We use the lightweight DAG search budget on both tasks: at most 30 search steps and up to 3 parents per new node. To isolate the contribution of OptMaster, we introduce no task-specific expert priors or handcrafted operators. Each node is evaluated with a parallelism degree of 3 on the computing cluster and every run uses a dedicated 32-core CPU worker. Implementations may use standard acceleration libraries (e.g., `Numba`) and minimal utility code for loading instances and computing objectives, so the LLM focuses on heuristic design rather than routine infrastructure code. We report the best objective value found within the budget.

### 4.2.2. MAXCUT

**Task and Evaluation.** Given an undirected graph $G = (V, E)$, MAXCUT seeks a binary assignment $x \in \{0, 1\}^{|V|}$ maximizing the number of edges crossing the cut,

$$\max_{x \in \{0,1\}^{|V|}} \sum_{(i,j) \in E} \mathbb{I}[x_i \neq x_j]. \qquad (17)$$

We evaluate on the Gset suite (Ye, 2003) and focus on Gset70 (10,000 nodes, 9,999 edges), a standard large-scale MAXCUT benchmark that remains challenging for both exact solvers and modern heuristics. We compare with representative classical and neural baselines, including BLS (Benlic & Hao, 2013), DSDP (Benson & Ye,

2008), PI-GNN (Schuetz et al., 2022), iSCO (Sun et al., 2023), MCPG (Chen et al., 2025a), L2A (Open-Finance-Lab, 2025), and Gurobi (1 hour time limit). To reduce implementation and environment variance, we report baseline numbers from the RLSolver (Open-Finance-Lab, 2025) benchmark under a unified protocol.

**Main Results.** Table 1 summarizes MAXCUT results on Gset70. OptMaster automatically discovers a multistart heuristic, which we refer to as Belief Propagation Guided Decimation (BPGD). The discovered procedure combines max-sum belief propagation for probabilistic guidance, adaptive decimation for incremental variable fixing, and a final greedy polishing stage, forming a fully executable algorithm rather than a one-off solution. Under a short runtime budget, the discovered heuristic attains a cut size of 9,587. When we increase the *runtime budget of the same heuristic* to 6 hours, it further improves to **9,590**. This scaling behavior indicates that OptMaster can quickly provide a strong algorithmic template, and that additional computation or expert iteration can build on the discovered heuristic to achieve further gains on large-scale NP-hard instances.

*Table 1.* MAXCUT results on Gset70 benchmark (10,000 nodes, 9,999 edges). Cut size reported (higher is better). All baseline results are taken from the RLSolver benchmark (Open-Finance-Lab, 2025). Best in **bold**.

| Method | Gset70 ↑ |
|---|---|
| PI-GNN (Schuetz et al., 2022) | 9421 |
| iSCO (Sun et al., 2023) | 9442 |
| DSDP (Benson & Ye, 2008) | 9456 |
| BLS (Benlic & Hao, 2013) | 9541 |
| MCPG (Chen et al., 2025a) | 9578 |
| Gurobi (1h) (Gurobi Optimization, LLC, 2024) | 9579 |
| L2A (Open-Finance-Lab, 2025) | 9586 |
| **OptMaster** (Discovered heuristic, default budget) | 9587 |
| **OptMaster** (Discovered heuristic, extended budget) | **9590** |

**Verifier diagnostic.** As detailed in Appendix B.3, the verifier plays a pivotal role by offering diagnostic reference points, such as rejecting candidates worse than trivial coloring baselines, thereby preventing the agent from pursuing dead-end heuristics.

### 4.2.3. CIRCLE PACKING

**Task and Evaluation.** We study circle packing in a unit square. Given $n$ circles with centers $(x_i, y_i) \in [0, 1]^2$ and radii $r_i \geq 0$, the goal is to maximize the sum of radii under containment and pairwise non-overlap constraints. The full mathematical program is provided in Appendix C.1. We report the best feasible objective value, verified by an independent checker.

We use the Packomania database (Friedman, 2025), which records best-known solutions for many $n$ while provably optimal solutions are unknown for most settings. Following recent LLM-based discovery work (Novikov et al., 2025), we use $n = 26$ as the primary target and evaluate transfer on $n = 27$ and $n = 32$. We compare against AlphaEvolve (Novikov et al., 2025), ShinkaEvolve (Lange et al., 2026) and FM Agent (Li et al., 2025).

*Table 2.* Circle Packing results on $n = 26$ and $n = 32$ (sum of radii; higher is better). Best-known (Packomania) refers to the best-known objective values recorded in the Packomania database (Friedman, 2025).

| Method | $n = 26$ ↑ | $n = 32$ ↑ |
|---|---|---|
| Best-known (Packomania) | 2.63490000 | 2.93600000 |
| AlphaEvolve (Novikov et al., 2025) | 2.63586275 | 2.93794000 |
| FM Agent (Li et al., 2025) | 2.63597400 | – |
| ShinkaEvolve (Lange et al., 2026) | 2.63598283 | – |
| **OptMaster** | **2.63598308** | **2.93957277** |

**Main Results.** Table 2 summarizes Circle Packing results. OptMaster discovers a fully executable heuristic without using any commercial solver. The discovered procedure combines diverse layout initialization, an adaptive trust-region Sequential Linear Programming (SLP) routine for joint radius and position refinement, and symmetry-breaking perturbations to escape local optima. Under the same discovery budget used throughout (**30** search steps), OptMaster identifies improved solutions on the primary target $n = 26$, achieving **2.63598308**. The same algorithm transfers by only changing the number of circles, reaching **2.93957277** on $n = 32$. On $n = 27$, OptMaster finds a qualitatively different packing pattern compared to the configuration recorded in the Packomania database (Friedman, 2025), while maintaining strong objective value (**2.68597868**). Figure 6 visualizes the discovered layouts, and Appendix C provides the full algorithm and verification details.

### 4.3. Ablation Study

**Impact of *Code as Verifier*.** We ablate *Code as Verifier* by removing execution-grounded verification from node scoring. Candidate code is still executed, but the evaluator scores nodes only from the problem statement and generated code (no solver re-checks or solution reviews). As shown in Table 3, accuracy drops on all formulation benchmarks: under *Two-Interpretations*, by 9.6 (IndustryOR), 1.0 (MamoComplex) and 3.2 (OptMATH) points; under *Base*, by 7.7, 2.3, and 4.4 points, respectively. The same ablation also degrades algorithmic search, reducing Gset70 from 9587 to 9529 and circle packing ($n=26$) from 2.63598308 to 2.62840584.

*Table 3.* Ablation of *Code as Verifier* on formulation intelligence (accuracy, %, mean $\pm$ std over 3 runs) and heuristic discovery (best objective, higher is better).

| Method | Formulation Intelligence (Acc. %) | | | Heuristic Discovery (Best Obj.) | |
|---|---|---|---|---|---|
| | IndustryOR | Mamo-Complex | OptMATH | MAXCUT (Gset70) | Circle Packing ($n{=}26$) |
| GPT-5 (single-pass) | 72.0 | 73.4 | 50.6 | 9379 | 2.50945491 |
| **OptMaster (Two-Interpretations)** | **87.3 $\pm$ 1.2** | **95.6 $\pm$ 0.4** | **66.9 $\pm$ 1.3** | – | – |
| w/o *Code as Verifier* | 77.7 $\pm$ 0.5 | 94.6 $\pm$ 0.4 | 63.7 $\pm$ 0.3 | – | – |
| **OptMaster (Base)** | **82.7 $\pm$ 1.2** | **80.1 $\pm$ 0.5** | **57.6 $\pm$ 1.0** | 9587 | 2.63598308 |
| w/o *Code as Verifier* | 75.0 $\pm$ 0.8 | 77.8 $\pm$ 0.4 | 53.2 $\pm$ 0.7 | 9529 | 2.62840584 |

**Impact of *DAG-Based Exploration* Topology.** We investigate whether the structural complexity of the DAG and the parent selection modes are necessary. Figure 3 shows that connectivity matters under the same budget. Multi-parent DAG expansion ($k{=}3$) achieves the best objectives on both tasks, while restricting connectivity ($k{=}1$) or forcing a chain consistently underperforms.

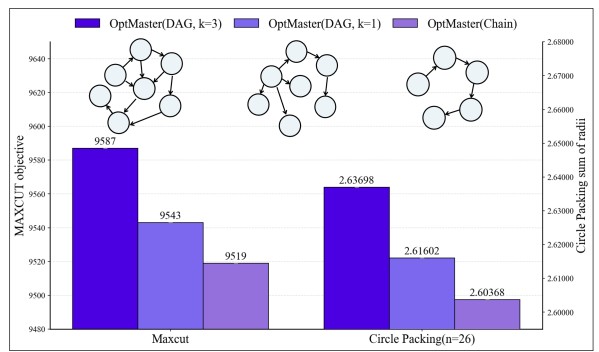

*Figure 3.* Effect of node connectivity in OptMaster on heuristic discovery. We compare DAG search with multi-parent fusion (DAG, $k{=}3$), a tree-structured restriction ($k{=}1$), and a linear chain. We report the best objective found under the same search budget on MAXCUT (Gset70) and Circle Packing ($n{=}26$).

### 4.4. Cost

In our setting, GPT-5 costs about $0.43 per formulation problem and about $5.08 per heuristic discovery task with a 30-node search budget.

## 5. Discussion

### 5.1. The OptMaster Paradigm

OptMaster performs strongly on both formulation and heuristic discovery under small budgets. On three formulation benchmarks, it consistently improves over the backbone model. On hard heuristic discovery tasks, it reaches competitive best-known solutions within a 30-step search and remains cost-efficient in our setting.

The gains follow three design choices. OptMaster explores multiple solution paths and delays commitment until evidence supports a path. *Code as Verifier* turns execution into a deterministic improvement signal. It recomputes objectives, checks feasibility, and reports targeted diagnostics. This computation-derived signal guides updates of both formulations and heuristics, and reduces hallucinated constraints and objectives. The multi-parent DAG makes successful edits reusable across paths by combining complementary improvements. The connectivity ablation supports this design, where $k{=}3$ outperforms $k{=}1$ and a chain under the same budget.

This paradigm changes the way we use LLMs for optimization. For OR modeling, it turns one-shot generation into a verifiable search process that produces solver-ready programs. For heuristic discovery, it outputs end-to-end executable heuristic discovery templates. These templates can improve with longer runtimes or expert refinement, while the core loop remains unchanged.

### 5.2. Limitations and Future Work

Industrial deployment adds challenges that are less visible in benchmarks. Data and constraints may be proprietary, objectives may evolve, and multi-stakeholder settings may not admit a single fixed objective. Another limitation is stochasticity. LLM-based search may follow different trajectories across runs, which is undesirable when strict repeatability is required. Still, many optimization workflows value finding a strong feasible solution at least once, and reproducibility can be improved through logging, stricter verification, and re-running promising branches.

OptMaster will be extended toward real-world optimization problems with noisy data, changing requirements, and tight budgets. Future work will also study how the explore-code-verify-learn loop generalizes beyond optimization to other executable, measurable tasks.

## Software and Data

We use publicly available benchmark instances and mathematics problems. Our framework and prompts are available at: https://github.com/lhhhappy/optmaster.

## Impact Statement

This paper introduces OptMaster, an LLM-based framework for optimization formulation and heuristic discovery. It aims to reduce human effort in building correct models and executable heuristics, which can benefit planning and resource allocation.

Potential risks include incorrect decisions due to modeling or verification errors, misuse for harmful objectives, and security risks from executing generated code. OptMaster mitigates part of these risks via execution-grounded verification and reproducible artifacts. For deployment, we recommend sandboxed execution, strict data and tool permissions, and human review before real decisions. Our experiments use public benchmarks and no personal data.

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

# A. Appendix

## A.1. System Overview

Figure 4 illustrates the system overview of the OptMaster framework. Given a natural language problem specification, OptMaster maintains a directed acyclic graph (DAG) of solution candidates and iteratively expands it through generation, execution, and verification. Each iteration selects one or more existing nodes as parents, composes or revises them into a new candidate solution, and evaluates it through executable verification. The resulting feedback is used both to score the solution and to guide subsequent exploration decisions. Unlike token-level refinement or fixed-depth search, OptMaster operates at the level of complete, executable solutions. All exploration is grounded in solver execution and verifier feedback, ensuring that progress is driven by objective improvements rather than textual plausibility.

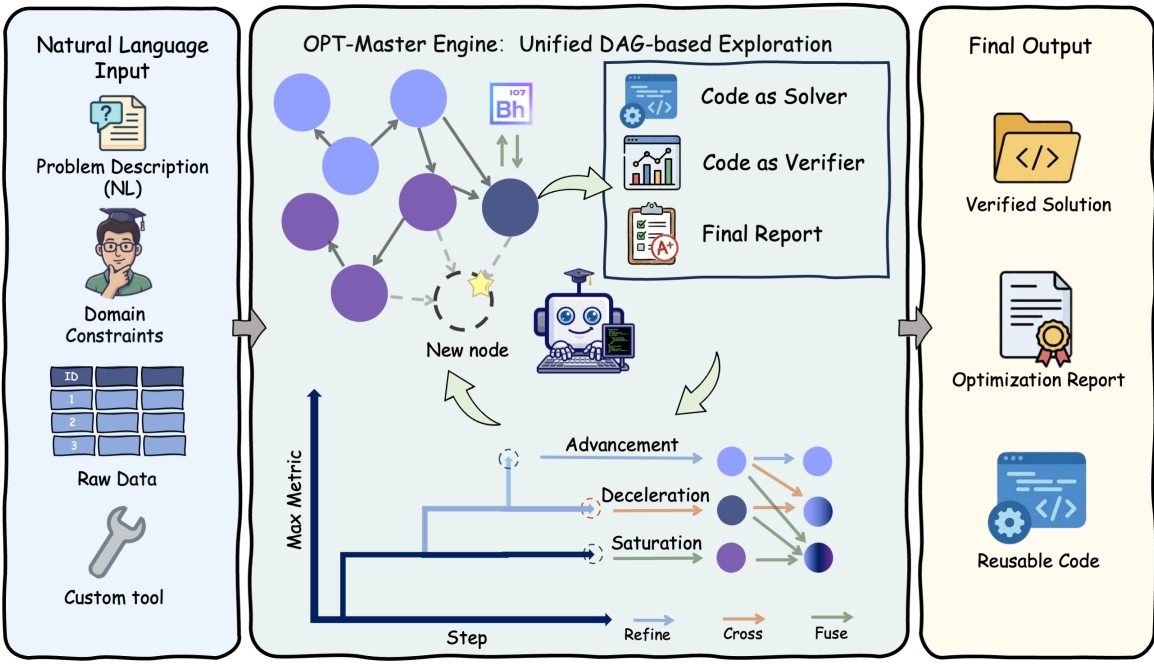

*Figure 4.* System overview of the OptMaster framework. It illustrates the end-to-end pipeline from natural language problem specification to verified optimization solutions, driven by a unified DAG-based exploration engine.

## A.2. Algorithm Walkthrough

Algorithm 1 presents the Stagnation-Responsive DAG Exploration strategy used in OptMaster. The algorithm maintains a pool of solution nodes $\mathcal{V}$ organized as a directed acyclic graph, where each node corresponds to a complete executable solution with an associated reward $r_v$ obtained from verification. At each step, the algorithm selects one or more parent nodes, generates a new candidate solution through composition or revision, and evaluates it via execution-grounded verification.

A key control signal is the stagnation state $S_t$, which quantifies how long the search has failed to improve the global best reward. Specifically, $S_t$ is computed from the number of steps $\tau_t$ since the last improvement and normalized by a window parameter $W$. As $\tau_t$ increases, $S_t$ grows linearly until it saturates at $1 - \varepsilon$, gradually shifting the search behavior from exploitation to exploration. When $S_t$ is small, the algorithm favors single-parent expansion and local refinement. As stagnation persists, larger parent sets are selected, enabling multi-parent fusion and encouraging structural exploration.

The function SeqInclude($S_t$) samples the number of parents $k \in \{1, 2, 3\}$. Parent selection explicitly balances solution quality and diversity. The first parent $p_1$ is sampled from the current node pool using a softmax distribution over rewards, ensuring that high-quality solutions are more likely to be expanded. Subsequent parents are selected to maximize a combined criterion of reward and distance from already selected parents. This distance-based term enforces population diversity and becomes increasingly influential as $S_t$ grows, preventing repeated refinement of similar solutions under stagnation. Here, we

---

**Algorithm 1** Stagnation-Responsive DAG Exploration

---

**Require:** Problem $\mathcal{I}$, Initial Node Pool $\mathcal{V}$, Buggy Set $\mathcal{B} = \varnothing$, Max Steps $T_{\max}$
**Ensure:** Best Solution $A^*$ and Best Node $v^*$

1: **for** $t = 1, \ldots, T_{\max}$ **do**
2:     $\mathcal{P}_t \leftarrow \varnothing$ {Initialize parent set}
3:     **if** $\mathcal{B} \neq \varnothing$ **then**
4:         $v \leftarrow \text{Pop}(\mathcal{B})$ {Repair Mode}
5:         $\mathcal{P}_t \leftarrow \{v\}$
6:     **else**
7:         *Update Stagnation State:*
8:         $r_t^* \leftarrow \max_{u \in \mathcal{V}} r_u; \quad \tau_t \leftarrow t - \max\{i \le t : r_i = r_t^*\}$
9:         $S_t \leftarrow \min(\tau_t/W, 1 - \varepsilon)$
10:      *Sample Expansion Mode:*
11:      $k \sim \text{SeqInclude}(S_t)$ where $k \in \{1, 2, 3\}$
12:      *Select Parents (Quality & Diversity):*
13:      Sample $p_1 \propto \exp(r_v/\beta)$ from $\mathcal{V}$
14:      $\mathcal{P}_t \leftarrow \{p_1\}$
15:      **for** $j = 2, \ldots, k$ **do**
16:         $p_j \leftarrow \arg\max_{v \notin \mathcal{P}_t} \left[ r_v + S_t \cdot \min_{p \in \mathcal{P}_t} d(v, p) \right]$
17:         $\mathcal{P}_t \leftarrow \mathcal{P}_t \cup \{p_j\}$
18:      **end for**
19:     **end if**
20:     *Execute Generation & Verification:*
21:     $\mathcal{T}_{\mathcal{P}} \leftarrow \{T_p \mid p \in \mathcal{P}_t\}$
22:     $(v_{\text{new}}, A_{\text{new}}, T_{\text{new}}, \text{status}) \leftarrow \text{NODELIFECYCLE}(\mathcal{I}, \mathcal{T}_{\mathcal{P}})$
23:     **if** status = FAIL **then**
24:         $\mathcal{B} \leftarrow \mathcal{B} \cup \{v_{\text{new}}\}$
25:     **else**
26:         $\mathcal{V} \leftarrow \mathcal{V} \cup \{v_{\text{new}}\}$
27:         **if** $r_{v_{\text{new}}} > r_{v^*}$ **then**
28:             $v^* \leftarrow v_{\text{new}}$
29:             $A^* \leftarrow A_{\text{new}}$
30:         **end if**
31:     **end if**
32: **end for**
33: **return** $A^*, v^*$

---

define the semantic distance $d(v, p)$ as the cosine distance between the plan embeddings of nodes $v$ and $p$. The embeddings are computed using a pre-trained model Sentence Transformers (Reimers & Gurevych, 2019), allowing the agent to penalize semantically similar parents even if their textual surface forms differ.

In addition to regular expansion, OptMaster maintains a separate *Buggy Set* that stores failed solution attempts. When this set is non-empty, the algorithm enters a repair mode and prioritizes revisiting failed nodes. Rather than discarding them, OptMaster treats such solutions as near-miss candidates and focuses on localized correction guided by verifier diagnostics. This separation improves sample efficiency by avoiding repeated rediscovery of similar failures.

## B. Case Study: MAXCUT on Gset70

The Maximum Cut (MAXCUT) problem is a core task in combinatorial optimization. It is widely used in both theory and practice. In statistical physics, MAXCUT corresponds to the ground state of an Ising spin glass (Zhang & Kamenev, 2025). In machine learning, it supports clustering and community detection in graphs (Diboune et al., 2024). It also appears in VLSI design and layout optimization (Rehfeldt et al., 2023). It is relevant to portfolio construction and related decision problems in quantitative finance (Chicano et al., 2025). We study the Gset70 instance, which has 10,000 nodes and 9,999

edges. The Gset70 instance is particularly challenging due to its large scale and sparse structure, making it a representative benchmark for evaluating whether OptMaster can autonomously discover nontrivial heuristics beyond local search.

### B.1. Search Trajectory on Gset70

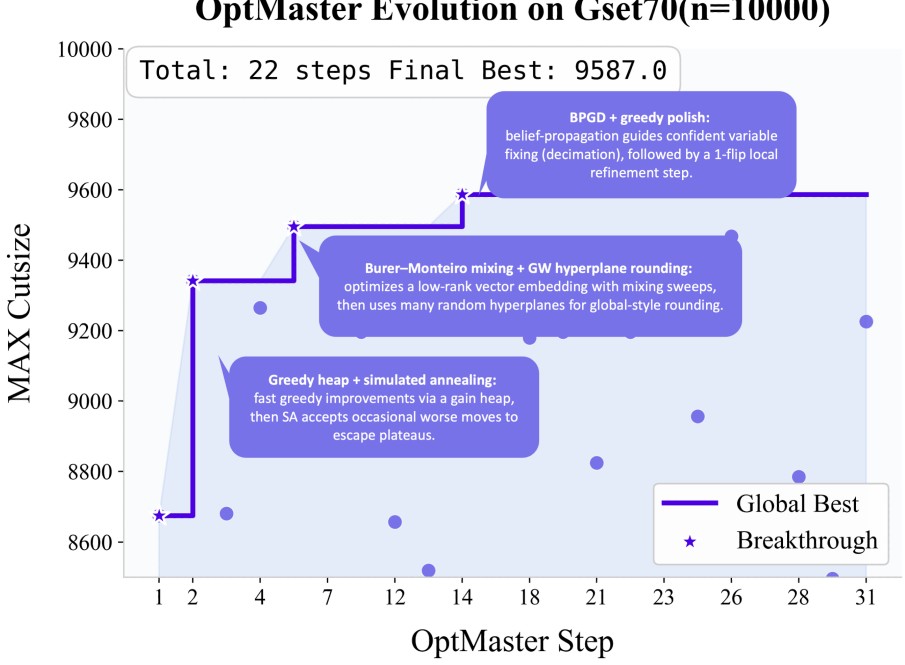

*Figure 5.* Search trajectory of OptMaster on Gset70. Star markers indicate breakthrough improvements.

Figure 5 shows the trajectory on Gset70. OptMaster reaches its final heuristic within **22** search steps. The cut size increases from 8675 to 9587. Star markers highlight four breakthrough steps with clear objective gains. Each breakthrough corresponds to a verifier-triggered structural modification of the heuristic, rather than stochastic fluctuation.

### B.2. Discovered Algorithm: Belief Propagation Guided Decimation (BPGD)

While each component is standard, the discovered heuristic differs from canonical BPGD variants in its decimation schedule, damping strategy, and verifier-driven restarts, which were not pre-specified but emerged through solution-level feedback.

OptMaster discovers a heuristic we call Belief Propagation Guided Decimation (BPGD). It combines probabilistic guidance with incremental variable fixing. The procedure has three phases.

**(1) Max-Sum BP.** It runs max-sum belief propagation on a factor graph with XOR potentials. It uses damping ($\alpha = 0.7$) for stability on loopy graphs.

**(2) Adaptive Decimation.** It fixes high-confidence variables in batches. The decimation rate starts at 5% and decreases to 1% as the graph shrinks. After each batch, it updates neighbor potentials and continues message passing.

**(3) Greedy Polish.** It applies iterative 1-flip improvements until no single flip improves the cut. It uses small random perturbations ($\epsilon = 0.05$) to break the $\mathbb{Z}_2$ symmetry during the process.

### B.3. Verifier Diagnostic and How It Guides Updates

OptMaster uses *Code as Verifier* to drive updates. The verifier recomputes the objective and checks feasibility. It also reports targeted diagnostics that describe the current failure mode. The discovered components are standard. The contribution is that OptMaster automatically composes them and tunes hyperparameters through solution-driven feedback.

The verifier provides more than a scalar score. It recomputes the cut, checks feasibility, and reports simple reference points. In early steps, one candidate achieved a cut of 8675, while a trivial 2-coloring baseline already reached 9196, so the heuristic was clearly underperforming. This gap directly triggered a shift toward stronger global exploration. The diagnostic also showed a high uncut-edge ratio of 0.134. After we introduced stronger global exploration via randomized restarts and perturbed BP initialization, the discovered heuristic reached 9587 and reduced the uncut-edge ratio to 0.041. The verifier then certified single-flip local optimality with max gain 0, suggesting that remaining gains should come from better exploration rather than a stronger local polisher.

For an intermediate solution $x$, the verifier checks 1-flip local optimality:

$$\forall i: \ f(x) \geq f(x^{(i)}), \tag{18}$$

where $x^{(i)}$ flips node $i$. In some steps, the verifier certifies 1-flip local optimality. It also finds an adjacent edge $(u, v) \in E$ where a simultaneous flip improves the cut:

$$f(x^{(u,v)}) - f(x) = 8. \tag{19}$$

This indicates that the search can stall under a single-node neighborhood. OptMaster then prioritizes cheap exploration moves, such as perturbed BP initialization and randomized restarts, to enter better basins before the final 1-flip polishing stage. This diagnostic distinguishes local optimality from global stagnation, allowing OptMaster to adapt its search paradigm instead of overfitting the local neighborhood.

## C. Case Study: Circle Packing in a Unit Square

Circle packing in a unit square is a long-standing open problem. It asks for the largest total radius of $n$ non-overlapping circles inside $[0, 1]^2$. Packomania records best-known solutions found over decades, but optimality is unknown for most $n$ (Friedman, 2025). We use $n = 26$ as the main target, following recent agent-based studies on this instance (Novikov et al., 2025). We also test transfer to $n = 27$ and $n = 32$.

### C.1. Problem and Benchmark

We maximize the sum of radii subject to containment and non-overlap:

$$\max \sum_{i=1}^{n} r_i \quad \text{s.t.} \quad \begin{cases} r_i \leq x_i \leq 1 - r_i, \ \ r_i \leq y_i \leq 1 - r_i & \forall i \\ (x_i - x_j)^2 + (y_i - y_j)^2 \geq (r_i + r_j)^2 & \forall i \neq j \end{cases} \tag{20}$$

where $(x_i, y_i)$ and $r_i$ are the center and radius of circle $i$. We evaluate by the best feasible objective value verified by an independent checker.

### C.2. Results Summary and Transfer

Table 2 summarizes the main results. On $n = 26$, OptMaster achieves **2.63598308**, improving over AlphaEvolve (2.63586275), ShinkaEvolve (2.63598283), and FM Agent (2.63597400). On $n = 32$, OptMaster reaches **2.93957277**. We also validate transfer on $n = 27$, where the same framework achieves **2.68597868**. Figure 6 visualizes the resulting layouts.

### C.3. Discovered Algorithm

OptMaster discovers an algorithm with three phases.

**(1) Diverse initialization.** It constructs multiple initial layouts from simple geometric templates, including hexagonal patterns, grid layouts, and jittered random placements. It then applies farthest-point sampling to select $n$ well-separated centers.

**(2) Adaptive trust-region SLP.** It alternates between (i) solving a linear program for radii and (ii) updating positions by linearizing the separation constraints around the current layout. A trust region controls step sizes. It expands after improvements and contracts after infeasibility.

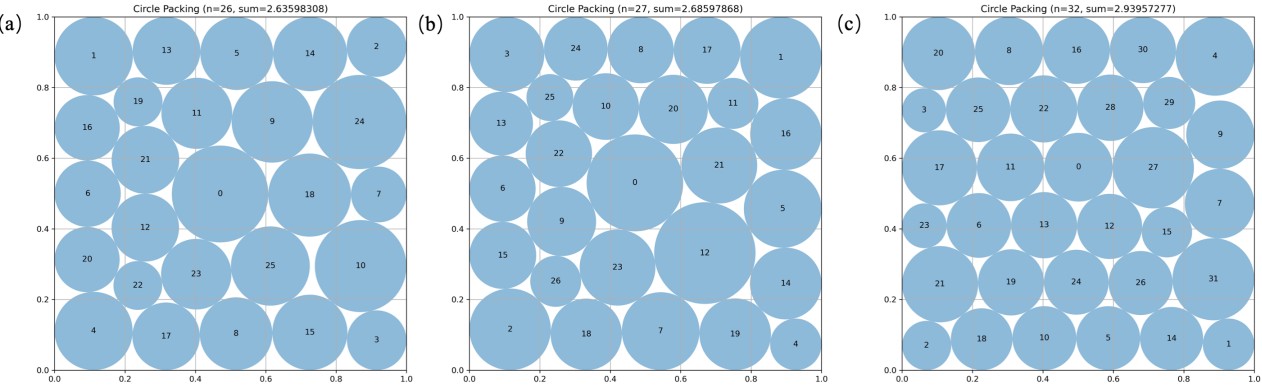

*Figure 6.* Solutions discovered by OptMaster for circle packing in a unit square. (a) $n = 26$, sum $= 2.63598308$. (b) $n = 27$, sum $= 2.68597868$. (c) $n = 32$, sum $= 2.93957277$.

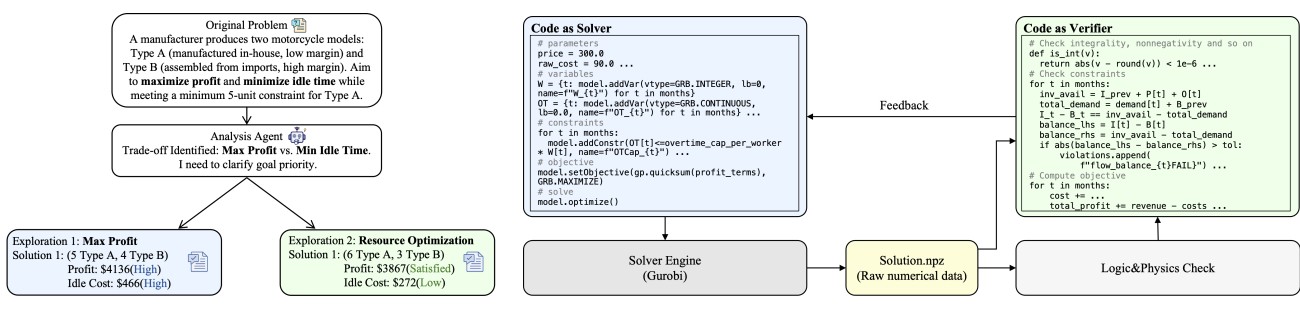

*(a)* Exploration Planning                *(b)* Code Decoupling Workflow

*Figure 7.* Formulation case studies. (a) Objective ambiguity triggers exploration of multiple candidate readings. Verification selects the formulation that matches the benchmark reference. (b) Code decoupling enables solution-level verification by recomputing objectives and checking constraints from raw numerical outputs.

**(3) Symmetry-breaking perturbations.** It applies small rotations and reflections to the best layout to generate new candidates. Each candidate is refined again by the SLP loop to escape local optima.

While each component is standard in continuous optimization, OptMaster discovers their combination, ordering, and adaptation purely from solution feedback, without domain-specific design.

### C.4. Verification Procedure

An independent verification routine checks boundary containment and pairwise non-overlap. If violations are detected, it solves a small linear program to shrink radii and restore feasibility before reporting the objective. This correction is conservative and can only decrease the objective, ensuring that reported values are strict lower bounds of the true solution quality.

## D. Case Study: Formulation Intelligence

This section highlights two frequent failure modes in formulation from text. The first is objective ambiguity. The second is silent modeling errors that look plausible in code. Figure 7 shows how OptMaster addresses both with exploration planning and execution-grounded verification.

### D.1. Exploration Planning for Objective Ambiguity

Figure 7(a) shows a motorcycle production problem with a hard constraint and two objectives. The prompt asks to maximize profit and minimize idle time. It does not specify priority or a trade-off rule. OptMaster treats this as an ambiguity signal. It explores two candidate readings. One prioritizes profit. The other prioritizes resource efficiency. Both solutions satisfy the constraint and are reasonable. Verification then selects the formulation whose solved objective matches the optimization

problem. This avoids early commitment and improves reliability when prompts admit multiple valid formalizations.

### D.2. Solution-level Verification via Code Decoupling

Figure 7(b) shows the verification pipeline. Code as Solver generates GurobiPy code and runs the solver. It saves only raw numerical outputs in `Solution.npz`. Code as Verifier is independent. It does not access the formulation code. It reads the original problem and `Solution.npz`. It checks feasibility, integrality, and key balance constraints. It also recomputes the objective from decision values. This decoupling prevents self-confirmation. It exposes errors such as missing constraints or incorrect objective terms. It also makes scoring reproducible, since it is driven by deterministic execution.

## E. Task Prompts

This appendix reports the task prompts used for heuristic discovery. The full framework prompt templates are provided in our code repository.

### E.1. Circle Packing in Unit Square ($n=26$)

Objective: Pack 26 non-overlapping circles inside the unit square $[0, 1] \times [0, 1]$ to maximize the sum of radii.
Variables: Circle $i$ has center $(x_i, y_i)$ and radius $r_i$.
Maximize: $\sum_{i=1}^{26} r_i$
Subject to:
1) Containment (for each $i$): $x_i - r_i \geq 0$, $x_i + r_i \leq 1$, $y_i - r_i \geq 0$, $y_i + r_i \leq 1$
2) Non-overlapping (for any $i \neq j$): $\sqrt{(x_i - x_j)^2 + (y_i - y_j)^2} \geq r_i + r_j$
3) Non-negative radii: $r_i \geq 0$
Note: The improvement is typically at the $10^{-6}$ level, so thorough optimization and breaking through bottlenecks are essential.

### E.2. MaxCut on Gset70

Objective: Partition nodes into two groups to maximize the number of cut edges.
Problem: Given an unweighted graph, assign each node $i$ a label $x_i \in \{0, 1\}$. An edge $(u, v)$ is cut if $x_u \neq x_v$. The cutsize is the number of cut edges.
Instance: Gset70 with 10,000 nodes and 9,999 edges (all weights are 1).
Constraints: Each node must be assigned to exactly one group, i.e., $x_i \in \{0, 1\}$ for all nodes.
Requirements: Use a deep exploration strategy, try multiple random seeds, keep intermediate output silent.

## F. Complete Formulation Results

Table 4 reports the complete formulation accuracy. Beyond the three main benchmarks reported in the main text, we additionally evaluate OptMaster on two further benchmarks, ComplexOR and LogiOR (the last two columns), where OptMaster (Two-Interpretations) reaches **100.0%** and **75.0%**.

*Table 4.* Formulation Intelligence accuracy (%) on three main and two additional benchmarks (mean ± std over 3 runs where applicable). Best results are in **bold**. "–" denotes results not reported by the original authors.

| Method | IndustryOR | Mamo-Complex | OptMATH | ComplexOR | LogiOR |
|---|---|---|---|---|---|
| OptiMUS (Ahmaditeshnizi et al., 2024) | 31.0 | 43.6 | 20.2 | 66.7 | – |
| SIRL-Qwen2.5-32B (Chen et al., 2025b) | 42.0 | 61.1 | 45.8 | – | – |
| DeepSeek-R1 | 45.0 | 67.9 | 40.4 | – | – |
| LLMOPT (Jiang et al., 2025) | 46.0 | 68.0 | 45.8 | 72.7 | – |
| SIRL-Qwen2.5-32B-COPT (Chen et al., 2025b) | 46.0 | 72.4 | 45.8 | – | – |
| ORThought (Yang et al., 2025) | 57.8 | – | – | 77.8 | 46.0 |
| OptiTree (Liu et al., 2025) | 54.0 | 81.5 | 52.4 | 84.2 | – |
| GPT-5 | 72.0 | 73.4 | 50.6 | 88.9 | 64.1 |
| OptMaster (Base) | 82.7 ± 1.2 | 80.1 ± 0.5 | 57.6 ± 1.0 | 94.4 ± 0.0 | 70.7 ± 1.1 |
| **OptMaster (Two-Interpretations)** | **87.3 ± 1.2** | **95.6 ± 0.4** | **66.9 ± 1.3** | **100.0 ± 0.0** | **75.0 ± 0.9** |

# G. Cross-Backbone Ablation

We replace the GPT-5 backbone with GPT-4o and DeepSeek-V3 and keep the rest of the system unchanged. Table 5 reports accuracy on the five formulation benchmarks.

*Table 5.* Accuracy (%) across foundation-model backbones on all five formulation benchmarks.

| Method | IndustryOR | Mamo-Complex | OptMATH | ComplexOR | LogiOR |
|---|---|---|---|---|---|
| GPT-4o | 39.0 | 52.7 | 24.7 | 83.3 | 41.3 |
| DeepSeek-V3 | 54.0 | 61.6 | 32.6 | 88.9 | 55.4 |
| GPT-5 | 72.0 | 73.4 | 50.6 | 88.9 | 64.1 |
| Ours + GPT-4o (Base) | 62.0 | 70.9 | 44.6 | 83.3 | 63.0 |
| Ours + GPT-4o (Two-Interpretations) | 67.0 | 91.1 | 52.4 | 88.9 | 68.5 |
| Ours + DeepSeek-V3 (Base) | 69.0 | 78.3 | 44.0 | 88.9 | 65.2 |
| Ours + DeepSeek-V3 (Two-Interpretations) | 73.0 | 94.6 | 54.8 | 100.0 | 68.5 |
| Ours + GPT-5 (Base) | 82.7 | 80.1 | 57.6 | 94.4 | 70.7 |
| **Ours + GPT-5 (Two-Interpretations)** | **87.3** | **95.6** | **66.9** | **100.0** | **75.0** |

