# OpenReview forum: "OptMaster: A DAG-Based Framework for Formulation and Heuristic Discovery in Optimization"
_ICML.cc/2026/Conference — ICML 2026 regular_

### Official Review · Reviewer_KQQN · 2026-03-07

**Soundness:** 3
**Presentation:** 2
**Significance:** 3
**Originality:** 3
**Overall Recommendation:** 4
**Confidence:** 3

**Summary:**

This paper proposes OptMaster, a unified optimization framework that integrates optimization problem formulation and heuristic discovery. The method organizes the search process using a directed acyclic graph (DAG) structure, enabling multi-parent inheritance to facilitate knowledge transfer across different search paths and improve solution diversity and quality. It also introduces a Code as Verifier mechanism that uses executable code for validation to reduce hallucinations in self-verification and improve solution reliability. Experimental results show that OptMaster achieves strong performance on multiple benchmarks, particularly in Formulation Intelligence and Heuristic Discovery tasks, where it outperforms existing methods and discovers high-quality heuristics under limited search budgets.

**Compliance With Llm Reviewing Policy:**

Affirmed.

**Final Justification:**

The author has addressed the issues I mentioned, so I have increased the score to 4 while maintaining the same level of confidence.

**Key Questions For Authors:**

1. What is the motivation for the proposed method?
2. For large-scale optimization tasks, how does OptMaster optimize the use of its computational resources?
3. How can hyperparameters (such as the number of search steps and the number of parent nodes) be adaptively adjusted to meet the requirements of different problems?

**Limitations:**

yes

**Strengths And Weaknesses:**

**Strengths:**
1. OptMaster proposes a new framework that integrates two tasks that are usually treated separately, optimization problem modeling and heuristic discovery, providing a unified solution.
2. OptMaster performs very well on multiple optimization benchmarks.
3. The proposed method demonstrates strong novelty.

**Weaknesses:**
1. The paper reads more like a technical report than an academic paper, as it lacks sufficient motivation.
2. OptMaster relies on a DAG-based search architecture, so computational and resource costs grow rapidly as the search space expands. For larger-scale or higher-dimensional problems, the memory and computational requirements may become prohibitive.
3. Although the paper mentions a “code-as-verifier” mechanism intended to avoid self-confirmation errors through execution-based validation, the model’s internal reasoning process may still lack sufficient transparency for many experts in real-world applications.

---

> ### Author Rebuttal · Authors · 2026-03-31
>
> ## 1.Response to W1&Q1: Motivation for the Proposed Method and Unified Framework
> We sincerely thank the reviewer for the profound insight. Real-world industrial optimization problems rarely fall strictly into pure formulation or heuristics. Tackling complex constraints typically demands Matheuristics—a hybrid paradigm that leverages exact mathematical modeling to simplify and bound subproblems, while simultaneously deploying heuristics for efficient exploration within the feasible region. While existing LLM-based methods treat formulation and heuristic discovery as isolated tracks, OptMaster is fundamentally designed to bridge this gap. By unifying both paradigms with the Directed Acyclic Graph (DAG) structrue, OptMaster allows formulation and heuristics to dynamically interact, and navigate the trade-off between mathematical rigor and search efficiency.
> The necessity of this unified paradigm are strongly evidenced by our Circle Packing experiment. Under the workflow, OptMaster did not merely output a standard heuristic but autonomously discovered a sophisticated hybrid strategy. It combined "local exact Sequential Linear Programming (SLP)" (derived from formulation insights for local refinement) with "global symmetry-breaking perturbations" (derived from heuristic exploration to escape local optima). This autonomous synthesis demonstrates that formulation and heuristics are not just parallel tasks in OptMaster, but cooperative tools that mutually enhance each other to solve high-dimensional spaces.
> ## 2.Response to W2&Q2: Resource Management and Scalability in Large-Scale Tasks.
> We thank the reviewer for this concern. However, we would like to clarify that OptMaster’s DAG does not suffer from exponential growth. Instead of blind expansion, OptMaster maintains a controllable search width. Algorithmically, the framework regulates DAG expansion using a stagnation-responsive mechanism that strictly limits the search breadth. At each step, the LLM-based agent performs a strategic pruning by selecting only the most promising 1 to 3 parent nodes for fusion, based on their performance and semantic diversity. This means the computational complexity grows linearly, not exponentially with the number of search iterations. This design ensures that memory cost and token costs remain strictly bounded even for high-dimensional problems.
> Our agent is capable of invoking external tools in real-time rather than generating code entirely from scratch. Both problem inputs and candidate solutions are provided to the LLM via persistent files, while summarized algorithmic experiences are efficiently transferred across different nodes, controlling LLM inference costs.
> Furthermore, on the engineering side, our framework is designed to offload heavy computational workloads to external high-performance computing clusters, allowing users to orchestrate massive-scale tasks.
> ## 3.Response to W3: Transparency for Real-World Experts
> While the LLM's internal reasoning is inherently a black box, OptMaster externalizes the optimization process to ensure transparency. The *Code as Verifier* mechanism produces a structured diagnostic report based on deterministic execution. Experts can directly inspect the independently generated verifier code itself, alongside the generated solver code, the raw numerical solutions, and the explicit diagnostics. Because the verification logic is written in plain code, it is fully auditable by human experts. Furthermore, the DAG acts as a fully interpretable trail of the entire decision-making process.
> ## 4.Response to Q3: Hyperparameter Adaptability and Framework Robustness
> We highly appreciate the reviewer’s insightful question regarding the adaptability of our hyperparameters. OptMaster is specifically designed  to meet the requirements of different problems rather than relying on manual tuning. The search dynamically terminates when OptMaster determines that an optimal strategy has been achieved and all promising multi-node fusion schemes have been exhausted. Regarding the number of parent nodes, this is dynamically regulated via our Stagnation-Responsive Expansion strategy. The system autonomously decides how many parents to fuse to balance exploration depth and breadth. Furthermore, our newly added sensitivity analysis in **Table R4 in our response to reviewer sKAZ** empirically demonstrates that setting a hard upper bound of k=3 parent nodes provides the optimal trade-off between search performance and computational consumption.
> Furthermore, OptMaster features a decoupled modular architecture. Therefore, to adapt to diverse unseen optimization tasks in the future, the core adaptive DAG search workflow remains entirely unchanged. Practitioners only need to update the problem descriptions and plug in domain-specific execution tools. This design ensures high scalability and robustness, allowing the framework to autonomously navigate new algorithmic search spaces.

---

> > ### Author Rebuttal · Reviewer_KQQN · 2026-04-01
> >
> > Thank you for the author's response. However, further clarification is needed:
> >
> > The mathematical model also changes during the iteration process. From a conventional understanding, a mathematical model is a mathematical representation of an optimization problem. If the mathematical model is changed, how can it be proven that the new model better represents the optimization problem?

---

> > > ### Author Response · Authors · 2026-04-01
> > >
> > > Thank you for this important follow-up. We agree that this question highlights an important distinction. In our view, “model change” means different things in the two settings, but both are grounded by a fixed evaluation mechanism.
> > >
> > > For formulation tasks, the optimization problem is fixed by the original natural-language specification. Changes during iteration are mainly corrections to an imperfect early formulation, for example, missing constraints, wrong objectives, or incorrect variable domains. Whether a revised formulation is better is not decided by the LLM itself. Our verifier is separated from the solver and recomputes feasibility and objective values directly from the original specification. So a revision is considered better only if it is verified to be more faithful to the same original problem.
> > >
> > > For heuristic discovery, the situation is different. The same problem (e.g., MAXCUT) can be approached by very different strategies, for example, spectral relaxation, belief propagation, or local search. These are genuinely different ways to solve the problem. However, they are all judged by the same fixed criterion, such as cut value for MAXCUT or sum of radii for Circle Packing. The verifier always checks this ground-truth metric, no matter which strategy produced the solution. So the solving strategy may change, but the evaluation standard does not.
> > >
> > > In short, the candidate model or strategy may change, but the criterion used to evaluate it remains fixed. We hope this clarification makes our intended distinction clearer.

---

### Official Review · Reviewer_VYKJ · 2026-03-13

**Soundness:** 2
**Presentation:** 3
**Significance:** 3
**Originality:** 3
**Overall Recommendation:** 3
**Confidence:** 4

**Summary:**

This paper proposes OptMaster, a DAG-based Large Language Model (LLM) framework designed to unify operations research (OR) modeling and heuristic discovery. Diverging from textual self-reflection, the framework utilizes a decoupled "Code as Verifier" mechanism to evaluate executable nodes and transfers structured semantic summaries across DAG nodes to enable multi-parent knowledge transfer. The proposed method achieves competitive results on both optimization formulation benchmarks and heuristic discovery tasks.

**Compliance With Llm Reviewing Policy:**

Affirmed.

**Key Questions For Authors:**

1. Can you provide compute- and search-budget-matched baseline results for both types of tasks to properly substantiate your claims regarding search efficiency and performance?
2. How sensitive are the formulation and heuristic tasks to the hyperparameters and modular choices within the overall search framework? Furthermore, how should the framework be adapted or evolved when facing diverse optimization tasks in the future?
3. Could you provide a concrete step-by-step example/trace demonstrating the DAG search process in action?

**Limitations:**

yes

**Strengths And Weaknesses:**

Strengths:
1. Unified and Practical System Design: Addressing both formulation and heuristic discovery within a single framework is highly challenging and valuable. The "Code as Verifier" mechanism elegantly decouples solution generation from objective evaluation, effectively grounding the LLMs in deterministic computation.
2. Effective Exploitation of Intrinsic Search Spaces: Both formulation and heuristic discovery tasks inherently possess underlying search spaces. The proposed DAG-based search strategy conducts efficient exploration within these intrinsic spaces, offering a promising pathway for iterative accuracy improvements.
Weaknesses:
1. Narrow Heuristic Evaluation and Unfair Baselines: The conclusions regarding heuristic discovery rely on a very limited set of instances (primarily MAXCUT Gset70 and Circle Packing for n=26/32). More critically, comparisons with baselines are not matched in terms of compute or time budgets (e.g., comparing directly against RLSolver's published historical best metrics rather than testing under controlled, identical conditions). This significantly weakens the claims of "high search efficiency".
2. Incomplete Experimental Validation: The current experiments lack evaluations on abstract modeling tasks, such as ComplexOR. Additionally, the ablation studies omit an analysis of how the DAG-based exploration topology affects the formulation tasks specifically.
3. While the framework integrates both formulation and heuristic discovery, the paper lacks an in-depth analysis of the necessity of placing these two distinct tasks within a single system. Furthermore, there is no discussion or empirical evidence demonstrating whether these two types of tasks mutually benefit or promote each other under the unified DAG search process.

---

> ### Author Rebuttal · Authors · 2026-03-31
>
> ## 1.Response to W1&Q1: Limited Evaluation and Computational Budget
> We have conducted additional experiments on a broader range of benchmarks, including Gset14, G22, G49, and G50 (as detailed in **Table R2 in our response to Reviewer 1hMw**). These results consistently demonstrate that OptMaster achieves SOTA performance across varying graph densities and scales.
> To address the concerns regarding computational cost, it is crucial to distinguish between the search time and the execution time. Our framework focuses on discovering better heuristic algorithms from scratch. Once discovered, the actual execution time of these heuristics is on the scale of seconds.
> To further clarify the search cost, as mentioned in the paper 4.2.1, OptMaster operates under a strictly lightweight DAG search budget of at most 30 search steps and up to 3 parents per new node. Under this setting, each node is evaluated with a parallelism degree of 3 on a 32-core CPU worker, a computational budget that is smaller than other methods requiring large-scale GPU parallelism. Furthermore, we have empirically measured the search overhead across diverse Gset instances. As detailed in Table R5, we present the number of explored nodes, the total token cost, and the average cost per node. Evaluating a single node takes only about 24 minutes on average and even as little as two minutes on simpler graphs (e.g., G49).
>
> **Table R5: Computational Cost on Diverse Gset Instances**
> | Exp | Nodes | Total Cost($) | Cost($)/Node |
> | :--- | :---: | :---: | :---: |
> | G14 | 6 | $1.25 | $0.21 |
> | G22 | 17 | $5.35 | $0.31 |
> | G49 | 27 | $1.19 | $0.04 |
> | G50 | 30 | $4.74 | $0.16 |
>
> Regarding the comparison, we must emphasize the paradigm shift in our approach. Many neural-based baselines (e.g., PI-GNN) require massive offline pre-training on GPU clusters and are often restricted to a single specific optimization problem. And for the formulation task, our nodes are restricted to a small number (at most three attempts per branch) to minimize redundant exploration. Under this setting, using GPT-5 costs only about $0.43 per formulation problem.
> ## 2.Response to W2: Incomplete Experimental Validation
> We thank the reviewer for the suggestion to include more modeling benchmark. Due to space constraints, We added detailed experiments in the **Response to W1 for Reviewer 1hMw**. And we reveal that the core challenge of modeling lies in precise semantic alignment between natural language and mathematical constraints, rather than the structural discovery of divergent algorithmic logic. Therefore, we intentionally limit the DAG node count in this phase to ensure minimal computational overhead. Instead, we quarantee the precision of modeling tasks through specialized mechanisms, including exploration planning, Code as Solver and Code as Verifier.
> ## 3.Response to W3: Necessity of the Unified Framework
> We sincerely thank the reviewer for this profound insight, due to space limitations, we have elaborated on this point in the **Response to W1&Q1 for Reviewer KQQN**.
> ## 4.Response to Q2: Hyperparameter Sensitivity and Future Task Adaptability
> We highly appreciate the reviewer’s insightful question, due to space limitations, we provide a comprehensive discussion in the **Response to Q3 for Reviewer KQQN**.
> ## 5.Response to Q3: DAG Visualization
> We thank the reviewer for the constructive suggestion. In the original Appendix B.1, we presented the search trajectory for the G70 instance. To further intuitively demonstrate the structural advantages of our DAG, we now provide a more detailed qualitative visualization of the DAG fusion strategy using the G50 instance.
> Specifically, when the search enters the Fuse Mode, the retrieval mechanism selects three parent nodes:
> - Parent A (SDP Relaxation, Score 5876): Employs Burer-Monteiro continuous relaxation with hyperplane rounding but the rounding process is prone to falling into local optima.
> - Parent B (Population Evolution, Score 5852): Incorporates a population memetic approach with path-relinking but the search scope is limited to disagreement nodes, restricting global exploration.
> - Parent C (Iterated Local Search, Score 5872): Utilizes 1-flip descent but the random perturbations are insufficient to escape deep local optima traps.
>
> Through the information integration inherent to the DAG, OptMaster accurately diagnosed the shared bottleneck across these three divergent paths: the inability to handle long-range constraints and execute non-local jumps. Consequently, it orchestrated a paradigm shift from discrete search to a physics-inspired approach, synthesizing a fundamentally novel heuristic that combines Belief Propagation, Reinforcement, and Decimation. By leveraging cavity fields to aggregate global constraints, this fused algorithm successfully guided the system across deep energy barriers that were insurmountable by standard 1-flip or continuous relaxations, ultimately achieving the Best Known Solution (5880).

---

> > ### Author Rebuttal · Reviewer_VYKJ · 2026-04-04
> >
> > Thanks for the authors' response, after reading the rebuttal so far, I think it is appropriate to maintain my current score.

---

> > > ### Author Response · Authors · 2026-04-08
> > >
> > > We sincerely thank you for confirming that your concerns have been fully resolved. We also greatly appreciate your constructive feedback throughout the review process, which has helped us significantly improve the paper.
> > > Building on the additional experiments included in our rebuttal, including broader Gset instances, multiple backbone LLMs, and new formulation benchmarks, we will incorporate all of these results into the revised manuscript. We also agree that an important next step is to move beyond benchmark evaluation and test OptMaster on real-world OR problems. If accepted, we will further expand the camera-ready version by adding more problem types for both the formulation and heuristic discovery settings.
> > > If there are any remaining concerns or reservations that we may have overlooked, we would be very grateful to hear them. We would be happy to provide further clarification or additional experiments if helpful.

---

### Official Review · Reviewer_sKAZ · 2026-03-14

**Soundness:** 2
**Presentation:** 2
**Significance:** 2
**Originality:** 2
**Overall Recommendation:** 4
**Confidence:** 3

**Summary:**

This paper proposes a DAG-based framework for LLM-driven optimization formulation and heuristic discovery. The main idea is to organize past solution attempts as nodes in a directed acyclic graph, where verified plans and summaries can be reused to guide future exploration. The method combines code generation, verification, and multi-parent expansion within a unified search framework. Empirically, the paper reports clear gains on optimization formulation benchmarks and competitive results on heuristic discovery tasks such as MAXCUT and circle packing.

**Compliance With Llm Reviewing Policy:**

Affirmed.

**Final Justification:**

The authors have addressed my main concerns, and I appreciate the additional clarifications and experiments. Accordingly, I am increasing my score to 4.

**Key Questions For Authors:**

1.	After a fused child node is created from multiple parent nodes, are the original parent nodes retained in the DAG, or can they be replaced or pruned? Is there any redundancy detection mechanism for removing obsolete nodes?

2.	When scoring nodes, are they evaluated independently? As the number of nodes grows, how does the method maintain a stable ranking without score drift? More generally, how are scores normalized when objective scales differ across tasks or datasets?

3.	Have the authors explored increasing the maximum number of parent nodes beyond three? It would be helpful to know whether using more parents further improves performance

**Limitations:**

yes

**Strengths And Weaknesses:**

Strengths:

1.	The paper introduces a DAG-based design for structuring and reusing past attempts, which is a reasonable and potentially useful way to organize search history and experience reuse.

2.	The proposed method shows clear improvements on optimization formulation benchmarks, suggesting that the framework is effective in this setting.

Weaknesses:

1.	The paper does not clearly explain whether the DAG is constructed per instance or shared across a dataset. It is also unclear how experience can transfer to new instances or datasets. Related generalization experiments are missing.

2.	The formulation experiments would be stronger with direct comparison to closely related intelligent modeling methods, such as OptiMUS [1] and OptiTree [2]. The related work section could also provide a more complete coverage of recent work in this direction.

3.	The computational cost of DAG construction is not sufficiently analyzed. Since the method involves iterative expansion, retrieval, and verification, the paper should report the corresponding inference or wall-clock overhead.

4.	On heuristic discovery, the gains on MAXCUT and circle packing are rather small. The paper also does not report repeated-run statistics such as standard deviation, making it difficult to judge whether the improvements are stable.

5.	The paper provides little qualitative analysis of the DAG itself. It would be helpful to visualize representative DAGs, show informative node contents, and illustrate concrete parent-fusion and retrieval examples.

[1] AhmadiTeshnizi, Ali, Wenzhi Gao, and Madeleine Udell. "Optimus: Scalable optimization modeling with (mi) lp solvers and large language models." arXiv preprint arXiv:2402.10172 (2024).

[2] Liu, Haoyang, et al. "Optitree: Hierarchical thoughts generation with tree search for LLM optimization modeling." Advances in Neural Information Processing Systems 38 (2025).

---

> ### Author Rebuttal · Authors · 2026-03-31
>
> ## 1.Response to W1: How the DAG is constructed and how experience can transfer?
> To clarify, the DAG is constructed per instance. For every specific optimization problem, a dedicated DAG is built and each node within this DAG represents a solution attempt. Experience and insights are transferred between these nodes within the same instance. For example, during heuristic discovery, different solution attempts fuse and learn from the execution feedback of prior nodes, eventually converging on a highly effective heuristic.
> ## 2.Response to W2: Direct comparisons and related work discussions in formulation tasks.
> We sincerely thank the reviewer for the suggestion. Given the strict page limits, please refer to **Response to W1 for Reviewer 1hMw** for detailed results.
> ## 3.Response to W3: Analyze the overhead of iterative DAG construction.
> We sincerely thank the reviewer for the question, due to space limitations, we elaborated on this point in the **Response to W1&Q1 for Reviewer VYKJ**.
> ## 4.Response to W4: Significance and Stability of Improvements
> Regarding the magnitude of improvements on tasks like Circle Packing, we respectfully clarify that a $10^{−6}$ level improvement is highly significant in this specific domain. More importantly, the core contribution of OptMaster is structural algorithm discovery rather than mere numerical tweaking. While the numerical gains might appear marginal, the actual heuristics and solution paths autonomously discovered are fundamentally different from prior methods. This demonstrates the framework's unique capacity to uncover genuinely novel algorithmic structures.
> Regarding the repeated-run statistics and standard deviation, we emphasize that OptMaster consistently converges to these SOTA-level solutions across multiple independent runs. This high stability is directly attributed to the inherent design of our DAG-based formulation. By iteratively fusing the algorithmic traits of multiple diverse parent nodes, the DAG topology effectively neutralizes the variance or bias introduced by initialization.
> ## 5.Response to W5: Qualitative Analysis and DAG Visualization
> We thank the reviewer for this suggestion. Due to space constraints, we have provided a detailed visualization of the DAG search process in our **Response to Q3 for Reviewer VYKJ**.
> ## 6.Response to Q1: Node Retention and Redundancy Pruning
> To address the concern of redundancy and maintain exploration diversity, our system incorporates a semantic clustering mechanism. Rather than arbitrarily discarding original parent nodes after merging, the system evaluates semantic distances and clusters highly similar nodes, effectively merging redundant evolutionary paths. Furthermore,  the computational efficiency does not degrade as the node count increases. First, the expansion complexity is strictly bounded because we only select 1 to 3 parent nodes for any merging operation. Second, the node selection process itself introduces negligible latency. The DAG operates within a manageable capacity limit, and the semantic representations required for distance calculations are cached immediately upon each node's creation. Consequently, the selection process merely requires lightweight, small-scale matrix operations, which imposes absolutely no computational bottleneck on the overall system performance.
> ## 7.Response to Q2: Scoring Stability and Normalization
> We appreciate the reviewer’s inquiry regarding the scoring consistency and stability of our system. In OptMaster, for heuristics tasks, nodes are indeed evaluated independently using a fixed, problem-specific evaluation function (e.g., an automated execution-grounded verification script or a specific objective value metric). For any given task, this evaluation standard serves as an objective that remains immutable through the search process. Because every node in the DAG is measured against the same static criterion, the absolute score of a node depends solely on its own performance, which inherently eliminates the risk of score drift as the node count increases. Each DAG explores solutions for only one specific problem. Nodes are evaluated independently using deterministic execution feedback specific to that problem.
> ## 8.Response to Q3: Ablation on Maximum Parent Nodes
> We thoroughly explored the impact of parent node quantities. In our current framework, we experimented with 1, 2, and 3 parent nodes, which is documented in our ablation studies. As shown in the Table R4, we use dmore parents and found that scaling up to 3 parents provides a strong balance between performance gains and context-window efficiency.
>
> **Table R4: Ablation on Maximum Parent Nodes in Gset70**
> | Exp | Nodes | Total Cost($) | Cost($)/Node | Best |
> | :--- | :---: | :---: | :---: | :---: |
> | k=1 | 30 | $4.37 | $0.15 | 9,519 |
> | k=2 | 29 | $7.00 | $0.24 | 9,476 |
> | k=3 | 22 | $3.10 | $0.14 | 9,587 |
> | k=4 | 29 | $6.93 | $0.24 | 9,588 |
> | k=5 | 24 | $7.79 | $0.32 | 9,522 |

---

> > ### Author Rebuttal · Reviewer_sKAZ · 2026-04-03
> >
> > Thank you for the detailed rebuttal. The authors have addressed most of my concerns. However, I still have some remaining concerns regarding W1.
> >
> > While the rebuttal clarifies that the DAG is constructed on a per-instance basis, this also highlights a practical limitation of the current framework. In realistic applications, one often needs to solve a family or stream of related instances within the same domain, rather than spending substantial search cost on each instance independently. From this perspective, instance-specific DAG construction may limit the practical value of the proposed experience-reuse mechanism, since the accumulated experience is only reused within a single instance.
> >
> > I would therefore appreciate further clarification on whether the DAG can be incrementally maintained or updated across related instances, and whether such cross-instance reuse could lead to measurable gains in efficiency or solution quality. Even a small experiment on sequential instances from the same problem class would strengthen the paper.

---

> > > ### Author Response · Authors · 2026-04-04
> > >
> > > We sincerely thank the reviewer for this insightful suggestion. We agree that cross-instance reuse is an important consideration for practical deployment. Inspired by this comment, we conducted a preliminary experiment on cross-instance reuse.
> > >
> > > We extracted experience from a completed search on G50 and injected it as expert priors into a new search on G70. The DAG logs in OptMaster make this straightforward: we can identify which strategies worked, which failed, and which directions were worth pursuing.
> > >
> > > From G50, we identified 8 key observations:
> > >
> > > **Findings:**
> > > (1) Burer-Monteiro relaxation with large-scale hyperplane rounding helps escape discrete local optima.
> > > (2) BP-guided decimation captures long-range graph structure and gives the strongest results.
> > > (3) ILS converges fast but plateaus early, so it is better as a baseline than as the main method.
> > >
> > > **Failures:**
> > > (4) Pure 1-flip local search gets stuck in deep local optima on sparse graphs.
> > > (5) SDP relaxation without diverse rounding leads to poor discrete solutions.
> > > (6) Memetic search underperforms continuous relaxation and message-passing methods on sparse graphs.
> > >
> > > **Promising directions:**
> > > (7) Start with Parallel Tempering (PT) or BP-guided decimation rather than local search.
> > > (8) Always add greedy 1-flip refinement at the end.
> > >
> > > We then ran G70 with these priors and compared the convergence trajectory against our previous from-scratch run. Results are shown below:
> > >
> > > | Setting | Step 5 | Step 10 | Step 14 | Step 22 |
> > > |---|---:|---:|---:|---:|
> > > | previous from-scratch run | 9196 | 9196 | 9328 | 9587 |
> > > | w/ transferred priors | 9539 | 9570 | 9588 | — |
> > >
> > > With transferred priors, G70 reaches 9588 at step 14, whereas the from-scratch run was still at 9328 at step 14. We will clarify this point and discuss cross-instance transfer more explicitly in the revision.

---

### Official Review · Reviewer_1hMw · 2026-03-17

**Soundness:** 2
**Presentation:** 3
**Significance:** 3
**Originality:** 2
**Overall Recommendation:** 4
**Confidence:** 4

**Summary:**

This paper proposes OptMaster, a unified LLM-based framework that addresses both optimization formulation and heuristic discovery within a single system. The core architecture organizes solution attempts as a Directed Acyclic Graph (DAG), where each node represents a complete executable attempt with a three-phase lifecycle: code generation for solving, independently generated code for verification, and summarization for knowledge transfer. The DAG structure enables multi-parent inheritance, allowing new nodes to combine insights from multiple prior attempts, with a stagnation-responsive mechanism that adaptively shifts from single-parent refinement to multi-parent fusion as search progress plateaus. A key design choice is "Code as Verifier," which replaces natural-language self-reflection with independently generated verification code that recomputes objectives and checks constraints deterministically, decoupled from the solver code. Using GPT-5 as the backbone, OptMaster is evaluated on three formulation benchmarks, achieving state-of-the-art accuracy, and on two heuristic discovery tasks, where it matches or surpasses best-known solutions within a 30-step search budget.

**Compliance With Llm Reviewing Policy:**

Affirmed.

**Final Justification:**

I am willing to raise my score, with the expectation that the camera-ready version (if accepted) will include: 1) a wider variety of problem types for each formulation and heuristic discovery setting; and 2) a preliminary investigation into combining both tasks within a unified framework — not necessarily as a full benchmark, but perhaps as one or two case studies with some discussion of the insights gained.

**Key Questions For Authors:**

See the weakness part.

**Limitations:**

See the Weakness part.

**Strengths And Weaknesses:**

### **Strengths**

- The paper is well-structured and easy to follow despite the system's complexity, with a logical progression from exploration planning to node lifecycle to DAG expansion.
- Key choices are grounded in identified limitations of prior work. Code as Verifier addresses unreliable natural-language self-reflection, stagnation-responsive expansion balances exploitation and exploration, and transferring summaries rather than raw code is a sensible abstraction for multi-parent inheritance.
- The figures communicate the framework clearly, particularly Figure 5's annotated search trajectory and Figure 7's concrete case studies that illustrate how the design choices work in practice.

### **Weakness**

The main concern with this paper is that the experimental evaluation is not sufficiently comprehensive or deep to support the breadth of the claims. While OptMaster is positioned as a unified framework spanning formulation and heuristic discovery, the experiments in both settings are narrow in terms of baselines, benchmarks, and configurations, making it difficult to fully assess the framework's generality and robustness. Specific issues are detailed below.

For formulation experiments, the comparisons are limited to frontier foundation models (GPT-5, DeepSeek-R1) and a few optimization-specific systems (SIRL, LLMOPT). Prompt-based methods, fine-tuning approaches, and experience-learning methods are absent, despite the related work section (§2) discussing several such approaches including OR-R1, StepORLM, and AutoFormulation. Additionally, established benchmarks such as LogiOR and OPTIBench are not considered, limiting the strength of the state-of-the-art claims.

For heuristic discovery, only a single MAXCUT instance (Gset70) is reported, despite the Gset suite containing many instances with varying sizes and densities. Gset70 has a notably unusual structure (10,000 nodes but only 9,999 edges, nearly a tree), and it is unclear why this particular instance was chosen. For Circle Packing, the evaluated cases (n=26, 27, 32) are near-optimal configurations where improvements occur at the 10⁻⁶ level (as acknowledged in Appendix E.1), representing refinement near convergence rather than discovery on harder, less-explored instances. Evaluation on problems with larger optimality gaps would better demonstrate the framework's value.

Across all experiments, only GPT-5 is used as the backbone LLM. The paper does not investigate how performance varies with different models, leaving it unclear whether the results reflect the framework's design or GPT-5's capabilities. Relatedly, despite claiming to be a "unified framework," formulation and heuristic discovery are evaluated entirely in isolation. No experiment demonstrates a pipeline that begins with formulation and transitions to heuristic discovery within the same DAG, which would substantiate the unification claim.

## Questions

Besides the concerns raised in the Weakness part, I have the following questions:

1. In heuristic discovery, practitioners often face a trade-off between solution quality and computation time. Some users may prefer a heuristic that delivers a reasonable solution quickly rather than the best possible solution under an extended budget. Does OptMaster provide any mechanism for users to specify time-quality preferences, or to bias the search toward discovering faster heuristics rather than purely optimizing objective value?

---

> ### Author Rebuttal · Authors · 2026-03-31
>
> ## 1.Response to W1: Lacks comparisons in formulation experiments.
> We strongly agree with your insight regarding formulation problems. In the revised Related Work, we will restructure to formally classify existing methods into three paradigms: prompting-based methods, fine-tuning methods, and empirical learning methods. In our initial submission, we strategically selected the three most challenging benchmarks in the field. However, we fully agree with your consideration regarding established benchmarks. Therefore, we have expanded our evaluation to include additional benchmarks. As shown in the Table R1, it demonstrates that OptMaster achieves SOTA performance on them.
>
> **Table R1: Accuracy comparisons on added benchmarks with more methods (%).**
> |Benchmark|IndustryOR|Mamo-Complex|OptMATH|ComplexOR|LogiOR|
> |------|---|---|---|---|---|
> |OptiMUS|31.0|43.6|20.2|66.7|-|
> |SIRL-Qwen2.5-32B|42.0|61.1|45.8|-|-|
> |LLMOPT|46.0|68.0|45.8|72.7|-|
> |ORThought|57.8|-|-|77.8|46.0|
> |OptiTree|54.0|81.5|52.4|84.2|-|
> |ours(Base)|82.7|80.1|57.6|94.4|70.7|
> |ours(Two-interpretions)|**87.3**|**95.6**|**66.9**|**100.0**|**75.0**|
> ## 2.Response to W2: Evaluates on narrow instances.
> Regarding the MAX-CUT task, we thank the reviewer for the sharp and insightful observation. We intentionally selected Gset70 because it represents a distinct and highly challenging category of OR problems. For optimization systems, handling the sheer scale of 10,000 variables is a challenging test that often breaks standard LLM reasoning.  However, we completely agree that evaluating only on a sparse tree-like structure does not capture the full diversity of the Gset suite. To provide a more comprehensive evaluation, we have now conducted additional experiments on G14, G22, G49 and G50, which feature much higher graph densities.
> As shown in the Table R2, OptMaster maintains its SOTA performance across the massive sparse topologies and the denser instances.
>
> **Table R2: Performance of OptMaster across diverse MAXCUT instances (Gset)**
> | Instance | G14 | G22 | G49 | G50 |
> | :--- | :---: | :---: | :---: | :---: |
> | Nodes | 800 | 2,000 | 3,000 | 3,000 |
> | Edges | 4,694 | 19,990 | 6,000 | 6,000 |
> | BLS | 3,064 | 13,359 | 6,000 | 5,880 |
> | OptMaster | **3,064** | **13,359** | **6,000** | **5,880** |
>
> Furthermore, we analyzed the heuristics it automatically discovered for these different instances. For example, OptMaster autonomously tailored distinct strategies based on problem structures: it discovered a probability-based message-passing method, Belief Propagation Decimation with Extremal Optimization polish for G22. Meanwhile, for instance G50, it synthesized a continuous relaxation approach utilizing Vector-spin SDP relaxation combined with randomized hyperplane rounding.
> Regarding the Circle Packing task, we have a detailed explanation in the **Response to W4 for Reviewer sKAZ**.
> ## 3.Response to W3: Relies on GPT-5 and the unified pipeline.
> We appreciate this critical observation. To isolate the framework's contribution from the backbone model, we have added comprehensive ablation studies using the DeepSeek V3 and GPT-4o. The results demonstrate that OptMaster provides consistent performance gains independent of the underlying LLM, demonstrating the effectiveness of our framework.
>
> **Table R3: Accuracy across different foundation models and benchmarks (%).**
> | Method | IndustryOR | Mamo-Complex | OptMATH | ComplexOR | LogiOR |
> | :--- | :---: | :---: | :---: | :---: | :---: |
> | GPT-4o | 39.0 | 52.7 | - | 83.3 | 41.3 |
> | DeepSeek-V3 | 54.0 | 61.6 | 32.6 | 88.9 | 55.4 |
> | GPT-5 | 72.0 | 73.4 | 50.6 | 88.9 | 64.1 |
> | Ours + GPT-4o (Base) | 62.0 | 70.9 | 44.6 | 83.3 | 63.0 |
> | Ours + GPT-4o (Two-Interpretations) | 67.0 | 91.1 | 52.4 | 88.9 | 68.5 |
> | Ours + DeepSeek-V3 (Base) | 69.0 | 78.3 | 44.0 | 88.9 | 65.2 |
> | Ours + DeepSeek-V3 (Two-Interpretations) | 73.0 | 94.6 | 54.8 | **100.0** | 68.5 |
> | Ours + GPT-5 (Base) | 82.7 | 80.1 | 57.6 | 94.4 | 70.7 |
> | Ours + GPT-5 (Two-Interpretations) | **87.3** | **95.6** | **66.9** | **100.0** | **75.0** |
>
> Regarding the "Unified Framework" claim, we apologize but due to space limits, we have provided a detailed explanation in our **Response to W1&Q1 in Reviewer KQQN**.
> ## 4.Response to Q1: Can users specify preferences?
> We sincerely thank the reviewer for highlighting the practical necessity of balancing solution quality and computational time in real-world heuristic discovery. To address this, OptMaster natively supports flexible quality-time trade-offs by allowing users to explicitly constrain the search space. Specifically, users can limit the maximum number of expanded nodes to enforce a faster, greedier search, or restrict the maximum search depth. Furthermore, users can impose direct wall-clock time budgets or introduce efficiency penalties into the execution-grounded reward function, directly biasing the system to discover faster, lightweight heuristics rather than solely pursuing the absolute optimal value.

---

> > ### Author Rebuttal · Reviewer_1hMw · 2026-04-01
> >
> > I thank the authors for their rebuttal. The performance of OptMaster is indeed impressive, and I appreciate the additional clarifications. That said, I still feel the experimental coverage could be broader. I am willing to raise my score, with the expectation that the camera-ready version (if accepted) will include: 1) a wider variety of problem types for each formulation and heuristic discovery setting; and 2) a preliminary investigation into combining both tasks within a unified framework — not necessarily as a full benchmark, but perhaps as one or two case studies with some discussion of the insights gained.

---

> > > ### Author Response · Authors · 2026-04-03
> > >
> > > We sincerely thank the reviewer for the constructive feedback. We agree that broader experimental coverage would further strengthen the paper, and we also believe that the next important step is to move beyond benchmark evaluation and test OptMaster on real-world OR problems. If accepted, we will expand the camera-ready version by adding more problem types for both the formulation and heuristic discovery settings.

---

### Decision · Program_Chairs · 2026-04-30

**Decision:**

Accept (regular)

**Comment:**

A DAG-based evolving framework is adopted to generate problem formulations and code within a unified framework. Reviewers positively commented on the novelty and technical soundness, noting the logical progression from exploration planning to node lifecycle management and DAG expansion. After the rebuttal, the remaining concerns are still centered on the limited scope of experiments, as raised by Reviewer VYKJ and Reviewer sKAZ, as well as the effects and insights related to the two tasks coupled within the unified framework, also noted by the both reviewers. The generalization to more diverse instances should be systematically evaluated rather than demonstrated on individual cases. I did not observe significant performance gains, as the improvements in the two simple optimization tasks appear marginal.